

# Dynamic changes in moisture content and applicability analysis of a typical litter prediction model in Yunnan Province

Yunlin Zhang and Lingling Tian

School of Biological Sciences, Guizhou Education University, Guiyang, Asian, China

## ABSTRACT

**Background:** Forest fire risk predictions are based on the most conservation daily predictions, and the lowest litter moisture content of each day is often used to predict the day's fire risk. Yunnan Province is the area with the most frequent and serious forest fires in China, but there is almost no research on the dynamic changes and model predictions of the litter moisture content in this area. Therefore, to reduce the occurrence of forest fires and improve the accuracy of forest fire risk predictions, it is necessary to understand these dynamic changes and establish an appropriate prediction model for the typical litter moisture content in Yunnan Province.

**Method:** During the fire prevention period, daily dynamic changes in the litter moisture content are obtained by monitoring the daily step size, and the relationships between the litter moisture content and meteorological elements are analyzed. In this study, the meteorological element regression method, moisture code method and direction estimation method are selected to establish litter moisture content prediction models, and the applicability of each model is analyzed.

**Results:** We found that dynamic changes in the litter moisture content have obvious lags compared with meteorological elements, and the litter moisture content is mainly related to the air temperature, relative humidity and wind speed. With an increase in the sampling interval of meteorological elements, the significances of these correlations first increase and then decrease. The moisture content value obtained by directly using the moisture code method in the Fire Weather Index (FWI) significantly different from the measured value, so this method is not applicable. The mean absolute error (MAE) and mean relative error (MRE) values obtained with the meteorological element regression method are 2.97% and 14.06%, those from the moisture code method are 3.27% and 14.07%, and those from the direct estimation method are 2.82% and 12.76%, respectively.

**Conclusions:** The direct estimation method has the lowest error and the strongest extrapolation ability; this method can meet the needs of daily fire forecasting. Therefore, it is feasible to use the direct estimation method to predict litter moisture contents in Yunnan Province.

Corresponding author
Yunlin Zhang,
zhangyunlin@gznc.edu.cn

## INTRODUCTION

The litter moisture content (LMC) represents the internal water content of litter and directly affects the difficulty with which litter is ignited as well as a series of fire behavior indicators after a fire occurs; accurate LMC values are of great significance for assessing forest fire risk (*Deeming & Brown, 1975*; *Bradshaw et al., 1984*; *Chuvieco, Agiado & Dimitrakopoulos, 2004*). The LMCs obtained by the drying method are the most accurate, but this method requires the drying of litter, so it cannot be used to obtain instantaneous LMCs and cannot be applied in practice (*Jin & Chen, 2012*). Therefore, it is necessary to determine the dynamic changes that occur in the LMC, analyze its influencing factors, and seek a high-precision LMC prediction method for use in forest fire risk predictions.

Yunnan Province is a high-incidence area of forest fires in China. According to statistics, there were 6,410 forest fires in the region in 2001–2017, accounting for approximately 20% of the number of forest fires in Southwest China, with an average annual burned area of 150,000 ha (*Zhang, Guo & Hu, 2021*). However, few studies have been conducted on the LMC in this area, and the dynamic changes that occur in different types of LMCs respond differently to the environment (*Byram & Jemison, 1943*; *Anederson, Schuette & Mutch, 1978*; *Wagner, 1979*; *Pech, 1989*; *Holden & Jolly, 2012*). Understanding the dynamic changes that occur in the typical litter moisture content in Yunnan Province during the fire prevention period and obtaining an LMC prediction model with a high accuracy and strong extrapolation ability are of great significance.

At present, LMC prediction methods mainly include remote sensing estimations, meteorological element regression methods, semiphysical methods and process-based methods. Among them, remote sensing estimation methods are suitable for large-scale LMC research; but its moisture content is calculated through the model obtained by ground monitoring moisture content, with the change of climate area, terrain and vegetation type, the accuracy of the method will be affected. Therefore, this method depends on the prediction model obtained by ground monitoring moisture content to a certain extent (*Toomey & Vierling, 2005*; *Nolan et al., 2016*). Process-based methods can reveal the dynamic change mechanisms of litter water, and the prediction effects of these methods are the best. However, due to the complexity of process-based models and the numerous required parameters, it is difficult to apply these methods in practice (*Matthews & Mccaw, 2006*; *Matthews et al., 2007*). Meteorological element regression methods predict the LMC by establishing correlations between the LMC and meteorological elements. These methods are simple, but because the dynamic changes in LMC have significant spatial heterogeneity, these methods have poor extrapolation abilities, and considerable manpower and material resource costs would be required to popularize these methods (*Pook & Gill, 1993*; *Ruiz et al., 2002*). The semiphysical methods take physical models as their main equations, but in the semiphysical methods, some of the parameters are obtained by statistical methods. With simple research methods and strong extrapolation abilities, semiphysical methods share the advantages of both the physical methods and statistical methods (*Kandya et al., 1998*; *Slijepcevic, Anderson & Matthews, 2013*).

Meteorological element regression methods and semiphysical methods are the two most widely used method types (*Zhang & Sun, 2020*). In particular, for the litter types in specific areas, as a simple statistical method, the meteorological element regression method is simple and practicable, and the prediction accuracy may higher than the semiphysical method or process-based method to a certain extent (*Matthews, Gould & McCaw, 2010*; *Sun, Yu & Jin, 2015*). However, for the prediction model of LMC in Yunnan, the meteorological element regression method has not been studied systematically, so it needs to be analyzed. Therefore, these two methods are selected in this study to analyze the prediction effect of the LMC.

The direct estimation method is the most widely used method among semiphysical methods; this method was proposed by *Catchpole et al. (2001)*. It mainly takes the water vapor exchange equation as the main body, and the parameters are obtained through experiments. In addition, the fine fuel moisture code in the Canadian Fire Risk Rating System reflects the dryness of litters; this code also adopts a semiphysical method, has a good extrapolation ability, and is widely used in globally (*Viney & Hatton, 1989*; *Wotton & Beverly, 2007*; *Anderson & Anderson, 2009*). However, because its parameters are obtained from jack pine and lodgepole pine in Canada, whether it is suitable for the typical litter in Yunnan Province still needs to be verified.

Research shows that the LMC monitoring step size has a certain impact on the accuracy of a prediction model, and shortening the step size will improve the LMC prediction accuracy (*Sun, Yu & Jin, 2015*; *Zhang & Sun, 2020*). However, for this study, the main purpose was to apply prediction models to obtain forest fire risk predictions, and forest fire risk predictions are based on the most conservation LMC predictions, which are usually published once a day. Understanding the daily dynamic changes in the LMC is usually the most important concern of forest fire managers and researchers (*Zhang, 2018*). Therefore, in this study, the dynamic changes in the LMC were analyzed with a daily step size.

## MATERIALS & METHODS

### Overview of the study area

The study area is located in Jindian Forest Park (25°05′55″ N, 102°50′3″ E), Kunming City, Yunnan Province, the overview map of the study area is shown in Fig. 1. This area belongs to the central part of the Yunnan-Guizhou Plateau in China, with an elevation of approximately 1,890 m. It is adjacent to Dianchi Lake in the south and is surrounded by mountains on the other three sides. The study area has a low-latitude, subtropical plateau mountain climate with an average annual temperature of approximately 15 °C, high solar radiation, obvious dry season and rainy season distributions, and an average annual precipitation of approximately 1,800 mm, with precipitation mainly concentrated from June to October. The common trees in the study area include *Pinus yunnanensis*, which accounts for approximately 70% of the forest area in this region, *Pinus armandii*, *Platycladus orientalis*, *Cyclobalanopsis glaucoides* and *Quercus acutissima*. The main shrubs are *Myrica rubra* and *Camellia pitardii*. The fire prevention period in the study area lasts from December 1 of the current year to June 15 of the next year.
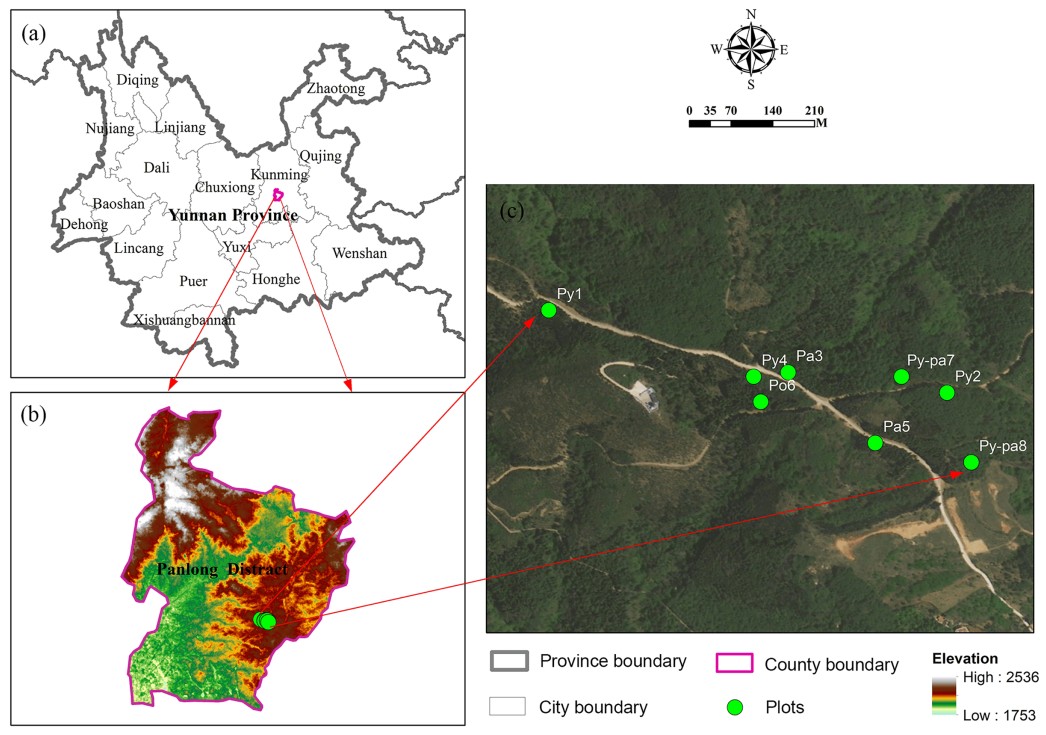

**Figure 1 Overview map of the study area.** (A) The administrative division map is reprinted from the National Geographic Information Resource Directory Service System (www.webmap.cn). (B) The 30 m resolution ASTER GDEMV2 data are from the Geospatial Data Cloud site, Computer Network Information Center, Chinese Academic of Sciences (http://www.gscloud.cn/sources/accessdata/421?pid=302). (C) The satellite image map data is from Map World released by National Platform for Common Geospatial Information Services (https://map.tianditu.gov.cn/).

## Sample plot setting

The typical forest types in the study area mainly include *Pinus yunnanensis* forests, *Pinus armandii* forests and *Platycladus orientalis* forests, and the slope, aspect and canopy density have significant effects on dynamic changes in the LMC (*Holden & Jolly, 2012*). Therefore, according to different slopes, aspects and canopy densities, this study selected *Pinus yunnanensis* forests, *Pinus armandii* forests, *Platycladus orientalis* forests and *Pinus yunnanensis-Pinus armandii* mixed forests as the research objects and selected eight forest stands with different aspects and slope positions for analysis. Among them, the forest type with distribution area are *Pinus armandii*, *Pinus yunnanensis*, *Pinus yunnanensis-Pinus armandii* mixed and *Platycladus orientalis* from high to low. Therefore, the number, slope and position of the stands are determined according to the area and location of the distribution. The stand is selected to reflect the actual research situation as much as possible, so it is selected randomly, and the thickness of fuelbed is also different.

Sample plots of 20 m × 20 m were set in each stand, the basic information and overview map of the stands is shown in Table 1 and Fig. 1 respectively. A total of eight samples plots are set, including 2, 3, 1 and two sample plots respectively for the forest stand of *Pinus*

**Table 1 The information of plot.**

| Plot | Forest type | Exposure | Slope (°) | Location | Depth of fuel (cm) | Fuel load (t·ha⁻¹) | Canopy density |
|------|-------------|----------|-----------|----------|--------------------|--------------------|----------------|
| 1 | Py1 | Semisouth | 10 | Upper | 3.9 | 2.83 | 0.5 |
| 2 | Py2 | South | 10 | Upper | 4.4 | 3.38 | 0.6 |
| 3 | Pa3 | North | 35 | Upper | 2.4 | 2.11 | 0.7 |
| 4 | Pa4 | South | 0 | Mid | 3.7 | 3.46 | 0.8 |
| 5 | Pa5 | South | 10 | Mid | 4.8 | 3.64 | 0.4 |
| 6 | Po6 | Semisouth | 30 | Mid | 1.6 | 4.81 | 0.8 |
| 7 | Py-pa7 | Seminorth | 20 | Upper | 3.4 | 4.62 | 0.5 |
| 8 | Py-pa8 | South | 25 | Mid | 3.5 | 4.22 | 0.4 |

**Note:**

To make the descriptions more concise, *Pinus yunnanensis*, *Pinus armandii*, *Platycladus orientalis* and *Pinus yunnanensis-Pinus armandii* are abbreviated as Py, Pa, Po and Py-pa, respectively. Sample plots one through eight are hereafter denoted as Py1, Py2, Pa3, Pa4, Pa5, Po6, Py-pa7 and Py-pa8.

*yunnanensis*, *Pinus armandii*, *Platycladus orientalis* and *Pinus yunnanensis-Pinus armandii* mixed. The monitoring time of the moisture content of the litters was 100 days, from February 10 to May 29, 2014.

## Monitoring the litter moisture content

In this study, only the leaves of litter on the surface were analyzed. Destructive sampling and weighting after drying is the LMC monitoring method that provides the closest estimation to the true value. Therefore, destructive sampling was selected in this study (*Zhang, 2018*). To prevent forest edge effects, all plots were set within areas 5–15 m from the forest edges. The main purpose of the daily LMC monitoring in this study is for applications in fire risk prediction; these prediction methods are based on the most conservative daily prediction. Therefore, the lowest daily LMC value (at 14:00) is selected for monitoring. During daily monitoring, three sampling points were randomly selected in each plot, and approximately 50 g litter was collected from each sampling point. The litter samples were put into envelopes and weighed immediately. The fresh weights were recorded, and then the samples were brought back to the laboratory. The litter was dried at 105 °C until the weight no longer changed, and this final weight was recorded as the dry weight. In the case of rainfall, the free water on the surface of the litter was wiped off with absorbent paper, and sealing pockets were placed in the envelopes to prevent the litter samples from wetting each other on the way back to the laboratory, as this would have affected the LMC determination results. The average daily LMC value of the three sample points in each plot was taken as the LMC value of the sample plot. The LMC calculation formula of each sample point is shown in Eq. (1):

$$M = \frac{W_H - W_D}{W_D} \times 100 \tag{1}$$

where M represents the LMC value (%); $W_H$ represents the wet weight of the litter (g); and $W_D$ represents the dry weight of the litter (g).

**Table 2 Basic overview of meteorological elements.**

|  | Temperature (°C) | Relative humidity (%) | Wind speed (m·s⁻¹) | Rainfall (mm) |
|---|---|---|---|---|
| Mean | 22.74 | 37.18 | 2.78 | 1.34 |
| Medium | 21.96 | 31.30 | 1.80 | 0.00 |
| Max | 32.33 | 92.30 | 9.97 | 42.81 |
| Min | 8.77 | 14.70 | 0.00 | 0.00 |

## Monitoring meteorological elements

Appropriate locations were selected in the study area, and HOBO Weather Stations (U30-2, Onset Company, US) were set up 1.5 m from the ground to monitor meteorological elements in the study area. The meteorological stations recorded meteorological elements such as the air temperature, relative humidity, wind speed and rainfall with a step length of 30 min. The monitoring time was earlier than the LMC monitoring time and was synchronized with the sampling time.

The basic meteorological element conditions in the study period are shown in Table 2. During the study period, the daily average air temperature was 22.74 °C, the highest and lowest daily air temperature was 32.33 and 8.77 °C, respectively. The variation range of daily relative humidity was 14.7–92.3%, the daily average relative humidity was 37.18%, and the daily relative humidity in half of the days during the monitoring period is lower than 31.30%. In addition, during the monitoring period, the daily average wind speed was 2.78 m·s⁻¹, the variation range of the wind speed was 0.00–9.97 m·s⁻¹, the daily maximum accumulate rainfall was 42.81 mm, and the daily average rainfall in rainfall days was 1.34 mm.

## Model description-multiple regression method of meteorological element

Dynamic changes in the LMC are mainly affected by the air temperature, relative humidity, wind speed and rainfall (*Glahn & Ruth, 2003*; *Australian Bureau of Meterology, 2012*). The increase of air temperature will improve the ability of air to carry moisture, which will increase the saturation humidity and decrease the relative humidity, the moisture of the litter will diffuse outward, and the LMC will decrease (*Alves et al., 2009*). The wind speed will accelerate the moisture diffusion of the litter and reduce the temperature of the litter, and also affect the dynamic change of LMC (*Fiorucci, Gaetani & Minciardi, 2008*; *Zhang et al., 2018*). In addition, rainfall will significantly increase litter moisture (*Wotton, 2009*). Other meteorological elements will also affect the dynamic changes of LMC, such as cloud, solar radiation and soil moisture content, but these elements have less and indirect impact on dynamic change of LMC than temperature, humidity, wind speed and rainfall (*Matthews, 2014*). Therefore, only these elements are selected in this study. And the responses of the LMC to these elements have certain lags (*Zhang & Sun, 2020*). Therefore, this study takes 14:00 as the node to calculate each average meteorological element $n$ days ago (except for rainfall, for which the accumulation at $n$ days before rainfall was measured), which was recorded as $M_{-n}$ ($M$ is air temperature, relative humidity, wind speed or rainfall, expressed by T, H, W and R, respectively; $n$ is 0–10.).

The LMC was taken as the dependent variable, and $M_{-n}$ was taken as the independent variable. The stepwise regression method was selected to obtain the appropriate predictors. The stepwise regression method is selected because it is a regression analysis method to establish the optimal regression equation. Since it introduces a new independent variable, it is necessary to test the old independent variables one by one, and eliminate the independent variables that are not significant in the partial regression sum of squares, so that they are introduced and eliminated until neither new variable are introduced nor old variables are deleted. After this step, the 'optimum' multiple linear regression equation is obtained, and the remaining independent variables are both important and there is no serious multicollinearity (*Zhang, 2018*). The model form of the regression method used for the meteorological elements is shown in Eq. (2):

$$M = b_0 + \sum_{i=0}^{10} X_i b_i \qquad (2)$$

where $M$ represents the LMC value; $b_0$ represents the prediction model constant; $X_i$ represents the meteorological element selected in the equation; and $b_i$ represents the coefficient.

## Model description-direct estimation method

The direct estimation method, proposed by Catchpole, is based on the equilibrium moisture content and directly uses the moisture content data of the previous day and meteorological data to predict the LMC (*Catchpole et al., 2001*). The main equation of this method is carried out by the water diffusion equation proposed by Byram in 1963 (*Byram & Nelson, 1963*), and the equation form is shown in Eq. (3):

$$\frac{dm}{dt} = \frac{M - E}{\tau} \qquad (3)$$

where $dm$ represents the value of the change in the LMC; $dt$ represents the time interval (h); $M$ represents the LMC value; $E$ represents the equilibrium moisture content; and $\tau$ represents the time lag (h).

The equilibrium moisture content models mainly include the Nelson model (*Nelson & Ralph, 1984*), Simard model (*Simard, Eenigenburg & Blank, 1984*), Van Wagner model (*Wagner, 1977*), etc.; compared with other equilibrium moisture models, the Nelson model is a semiphysical model that is based on the Gibbs equation, which has the advantages of both physical method and statistical method. The method is simple and has good extrapolation effect, and it is the most widely used. Therefore, in Eq. (3) of this study, the Nelson model is selected as the equilibrium moisture content model, and the model form is shown in Eq. (4):

$$E = \alpha + \beta log \left( -\frac{RT}{m} logH \right) \qquad (4)$$

where $R$ represents the universality constant, 8.314 J·K$^{-1}$·mol$^{-1}$; $T$ represents the air temperature (K); $m$ represents the atomic mass of water molecules, 18.0153 g·mol$^{-1}$; $H$ represents the relative humidity (%); and $\alpha$ and $\beta$ represent the parameters to be estimated.

In the direct estimation method, the time lag of litter must be fixed, and this study is carried out using a daily time step, so $\Delta t = 24\ h$. By discretizing Eq. (3), the discretized water diffusion equation expressed in Eq. (5) is obtained:

$$M_i = \lambda^2 M_{i-1} + \lambda(1-\lambda)E_{i-1} + (1-\lambda)E_i \tag{5}$$

where $M_i$ and $M_{i-1}$ represent the moisture content of the current day and the previous day (%), respectively; $E_i$ and $E_{i-1}$ represent the equilibrium moisture content of the current day and the previous day (%), respectively; and $\lambda$ represents the parameter to be evaluated in the model.

According to Eq. (4), the daily equilibrium moisture content is obtained in Eq. (5), the nonlinear estimation is carried out with the objective of minimizing the square sum of the measured and predicted LMC values, and the parameters $\lambda$, $\alpha$ and $\beta$ are obtained.

## Model description-moisture code method

The fine fuel moisture code (FFMC) in the fire weather index (FWI) represents the moisture content of fine fuel with a surface thickness of 1.2 cm and a load of 0.25 kg·m$^{-2}$; this value characterize the humidity of litter to a certain extent. The scale model is mainly used to convert values between the LMC and moisture code. At present, the FF scale model is the most commonly used model (*Wagner, 1987*), as shown in Eq. (6):

$$M_{FF} = 147.27 \frac{(101-F)}{(59.5+F)} \tag{6}$$

where $M_{FF}$ represents the predicted value of moisture content (%) and $F$ represents the FFMC value, which is the same as that described below.

In addition, since the study area is a dry and hot area during the fire prevention period, this study also adds an FX scale model that is suitable for dry and hot areas (*Lawson, Armitage & Hoskins, 1996*), as shown in Eq. (7).

$$M_{FX} = 32.87 \frac{(101-F)}{(13.28+F)} \tag{7}$$

$M_{FF}$ and $M_{FX}$ were compared with the measured LMC values to explore whether the direct scale model used was applicable for predicting moisture contents in this area. If it was not applicable, we would try to establish a correlation between the measured LMC value and the predicted $M_{FF}$ ($M_{FX}$) value and analyze the prediction effect of the model. Due to the large moisture content dataset, the variation in this series was larger than that in the predicted values; the natural logarithm was used to transform the data to realize variance uniformity. Then, a simple linear relationship was established between the measured values and the predicted values after the data was transformed (*Wotton & Beverly, 2007*), and an LMC regression prediction model was obtained based on the

moisture code. This model was recorded as the moisture code regression model, and the model form is shown in Eq. (8):

$$M = \alpha M_{FF(FX)}^{\beta} \tag{8}$$

where $M$ represents the measured LMC value (%); $M_{FF/FX}$ represents the moisture content values predicted based on the FF or FX scale models; and $\alpha$ and $\beta$ represent the parameters to be estimated.

## Statistical analysis

### Data process

It is generally believed that when the LMC value is higher than 35%, it is difficult for forest fires to occur (*Luke & Mcarthur, 1978*). Therefore, this study mainly carried out data analysis when the LMC value was lower than 35%. The basic statistics of the LMC data of eight sample plots were obtained, including the minimum value, maximum value, average value and other indicators.

### Correlation analysis

According to the meteorological data $M_{-n}$ sorted in section **Model description-multiple regression method of meteorological element**, the Spearman method was used to analyze the correlations between the LMC and meteorological elements. Taking the time series as the abscissa and each correlation coefficient as the ordinate, broken line charts of the correlation coefficients of eight typical LMC dynamic changes and meteorological elements with time were given.

### Model accuracy analysis

The FF and FX scale models were directly used to obtain the predicted LMC values, *t*-tests were performed with the measured values to analyze whether significant differences existed, and the average absolute error (MAE), relative average error (MRE) and root mean square error (RMSE) of the prediction model were calculated. The specific equations are shown as (9)–(11), and the suitability of directly using FF and FX scale models to predict the LMC was analyzed:

$$MAE = \frac{1}{n} \sum_{i=1}^{n} \left| M_i - M_j \right| \tag{9}$$

$$MRE = \frac{1}{n} \sum_{i}^{n} \frac{\left| M_i - M_j \right|}{M_i} \times 100\% \tag{10}$$

$$RMSE = \sqrt{\frac{1}{n} \sum_{1}^{n} \left( M_i - M_j \right)^2} \tag{11}$$

where $M_i$ represents the measured LMC value (%) and $M_j$ represents the predicted LMC value (%).

For all LMC prediction models established in this research, N-fold cross-validation was selected to test the model accuracies, the model errors were calculated according to Eqs. (9)–(11), and the prediction accuracy of the LMC prediction model was compared.

Taking the monitoring time as the abscissa and the measured and predicted values from the multiple regression method of meteorological element, the direct estimation method and the moisture content method as the ordinate, the dynamic changes in the moisture content during the study period were drawn, and the different changes in the LMC predicted by the three methods were compared. Since this study only analyzes the data with LMC below 35%, and the abscissa is the whole research time, the line segment of the figure will be discontinuous. In addition, taking the measured value as the abscissa and the predicted values from the three prediction methods as the ordinate, a curve was fit, and a 1:1 line was drawn to compare the prediction effects of the three methods in different LMC intervals.

### Model extrapolation comparison

The three LMC prediction models were obtained for eight plots, and the prediction methods were substituted into the remaining seven plots to obtain the prediction errors of the plot models when extrapolated to other plots. The minimum, maximum, mean and coefficient of variation in the errors were summarized and compared after the model extrapolation, and then these characteristic values were compared to analyze the stability of the extrapolation of the three LMC prediction models and the applicability of each model.

## RESULTS

### Litter moisture content statistical analysis

Table 3 shows the basic LMC conditions in eight sample plots. During the fire prevention period, the minimum number of days when the typical LMC in Yunnan was less than 35% occurred in Pa3, at 67 days, and the maximum occurred in Py2, reaching 85 days. The average LMC of Py1 was 20.75%, ranging from 12.83% to 34.00%. The average LMC of Py2 was slightly lower than that of Py1, but the range of variation was slightly larger in Py2 than in Py1. For litter of *Pinus armandii*, the average LMC value and variation range were largest in Pa5; Pa3 was the second largest, and Pa4 was the smallest. The average LMC of Po6 was 21.44%, the minimum was 14.82%, and the maximum was 33.55%. The LMC variation ranges of Py-Pa7 and Py-Pa8 were basically between those of Py and Pa. In the 8 plots, the mean LMC values were, from high to low, Py-Pa8, Po6, Py2, Py1, Py-Pa7, Pa4, Pa3 and Pa5, the minimum LMC value was recorded at Py2, at only 11.41%, and the maximum value was observed in Pa5, at 34.92%.

### Correlation analysis

All LMCs had significant negative correlations with the air temperature, and as the distance from the sampling time increased, the correlation first increased and then decreased. The LMC of Py1 had the weakest correlation with the air temperature; this LMC was only related to the air temperature of the previous 3–7 days, while the LMC of Pa4 had

**Table 3 Basic information on the moisture contents of the litters in the eight sample plots.**

| Forest type | N | Mean | Minimum | Maximum | Percentile 25 | Percentile 75 | Std. |
|---|---|---|---|---|---|---|---|
| Py1 | 82 | 20.75187 | 12.83160 | 34.00336 | 16.94826 | 23.18630 | 4.657846 |
| Py2 | 85 | 19.47853 | 11.40590 | 34.26783 | 15.76367 | 21.31457 | 5.354328 |
| Pa3 | 67 | 24.62457 | 17.79000 | 33.97879 | 21.28038 | 26.63266 | 4.055139 |
| Pa4 | 74 | 24.18342 | 17.30228 | 32.53193 | 20.86533 | 27.07264 | 4.073415 |
| Pa5 | 73 | 25.80851 | 15.64700 | 34.92262 | 20.62256 | 30.32734 | 5.590288 |
| Po6 | 81 | 19.94026 | 12.70799 | 34.68168 | 16.65793 | 22.74278 | 4.672823 |
| Py-pa7 | 79 | 21.43997 | 14.81812 | 33.54526 | 18.45807 | 23.69726 | 4.181648 |
| Py-pa8 | 84 | 19.09674 | 11.96713 | 31.30165 | 16.13363 | 21.50934 | 4.216814 |

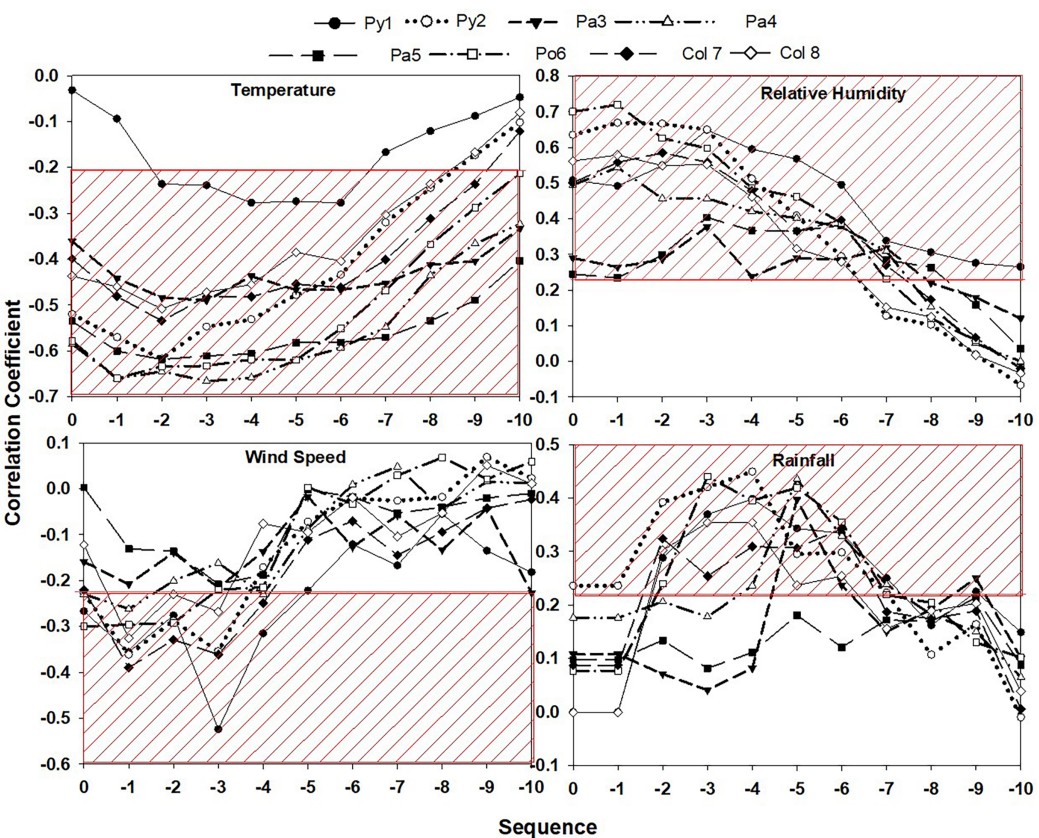

**Figure 2 Correlation coefficient between dynamic changes in the LMC and the meteorological factors of the last *n* days.** The red areas indicate that the LMC was significantly related to the meteorological factors at 0.05 level.

the strongest correlation with air temperature. The average relative humidities of the current day and the previous 6 days were significantly positively correlated with the LMC, and the LMC had the strongest correlation with the average relative humidity of the previous 3 days (Fig. 2).

## Applicability analysis of the direct use of scale models

Table 4 shows the $t$-test results for the predicted and measured values after the two scale models were directly used, as well as the model prediction errors. The $t$-test results of the measured LMC values and the values predicted by the FF (FX) scale model for the eight plots were almost all less than 0.01, and extremely significant differences existed between the measured and predicted values. For the two scale models, the minimum MAE value was 9.77%, and the maximum was 15.89%. The MRE and RMSE ranges were 47.80–64.53% and 11.94–19.07%, respectively. The prediction errors were large, and these models cannot be applied in practice. Therefore, it is not applicable to directly use the moisture code method to predict the moisture content of the typical litter in Kunming, Yunnan Province.

## Models

### Meteorological elements regression model

In the regression models of the eight sample plots, only the air temperature and relative humidity were selected as predictors. Most of the models selected the air temperature or relative humidity values of the previous day or 2 days. Only the Pa3 prediction model selected the average temperature of the previous 3 days as the predictor variable.
The variation ranges of the MAE, MRE, and RMSE values of the meteorological element regression models used to predict the LMC in the eight plots were 2.39–4.39%, 10.03–23.57%, and 2.98–5.43%, respectively. The MAE of Po6 was the lowest, and the MAE of Py-pa8 was the highest. The largest deviation between the measured and predicted LMC value was found in Py-pa8, and the smallest was found in Pa4 (Table 4).

### Moisture code regression model

To reduce the variability in the data, natural logarithms were taken for the measured and the predicted values of the scale model based on FF (FX), and a linear model between the two values was established. Table 4 gives the prediction results of the eight litter moisture contents obtained using the moisture code regression model. No significant difference was observed between the MAE and MRE values obtained by the FF-scale model and the FX-scale model for any litter type ($t$-test: $p > 0.05$). The MAE variation range of the moisture content prediction models obtained for the 8 plots based on the FF model was 2.50–5.09%, the minimum MRE was 12.22%, the maximum was 18.61%, and the RMSE variation range was 3.47–6.09%; based on the FX model, the MAE, MRE and RMSE values of the moisture content prediction models varied in ranges of 2.46–5.08%, 12.15–18.60% and 3.45–6.12%, respectively. The $R^2$ values of Py2, Po6, and Py-pa8 all exceeded 0.6, explaining the model well, and the prediction errors were smaller in these plots than those of the other plots. The prediction effect of the Pa understory LMC in the three plots obtained using the moisture code method was relatively poor. The MRE had a limit of 15% (Jin & Yan, 2012), and the LMC prediction effects of Pa3 and Pa5 cannot be applied to forest fire risks (Table 4).

**Table 4 Estimated parameters and errors of all the models established directly using the moisture code and three different methods.**

| Forest type | Method | | | | | | | | | | | | | | | | | | | | | | | | | | |
|---|---|---|---|---|---|---|---|---|---|---|---|---|---|---|---|---|---|---|---|---|---|---|---|---|---|---|---|
| | Directly using the moisture code | | | | | | Moisture code regression method | | | | | | | Multiple regression method of meteorological element | | | | | | Direct estimation method | | | | | | |
| | Scale model | t-value | $P$ | MAE (%) | MRE (%) | RM SE (%) | Sadle model | $\alpha$ | $\beta$ | $R^2$ | MAE (%) | MRE (%) | RM SE (%) | $b_0$ | $b_i$ | $X_i$ | MAE (%) | MRE (%) | RM SE (%) | $\alpha$ | $\beta$ | $\lambda$ | $\tau$ | MAE (%) | MRE (%) | RM SE (%) |
| Py1 | FF | 5.69 | 0.00 | 11.15 | 52.06 | 13.07 | FF | 10.88 | 0.27 | 0.35 | 2.81 | 13.84 | 3.61 | 10.847 | 0.188 | $H_{-1}$ | 2.78 | 13.80 | 3.87 | 28.57 | −4.83 | 0.41 | 13.54 | 2.79 | 13.81 | 3.57 |
| | FX | 3.71 | 0.00 | 11.53 | 52.97 | 15.25 | FX | 11.15 | 0.25 | 0.38 | 2.86 | 14.58 | 3.64 | | | | | | | | | | | | | |
| Py2 | FF | 3.31 | 0.00 | 9.77 | 47.80 | 12.91 | FF | 7.21 | 0.39 | 0.70 | 2.82 | 13.71 | 3.57 | 5.901 | 0.26 | $H_{-1}$ | 2.84 | 15.06 | 3.51 | 37.70 | −10.99 | 0.50 | 17.06 | 2.46 | 13.01 | 3.17 |
| | FX | 2.14 | 0.03 | 10.20 | 48.65 | 15.13 | FX | 7.40 | 0.37 | 0.70 | 2.76 | 13.47 | 3.57 | | | | | | | | | | | | | |
| Pa3 | FF | 12.23 | 0.00 | 15.83 | 64.18 | 16.94 | FF | 12.12 | 0.25 | 0.27 | 4.19 | 15.87 | 5.28 | 34.117 | −0.567 | $T_{-3}$ | 2.80 | 11.42 | 3.52 | 23.19 | 0.56 | 0.36 | 11.78 | 3.04 | 12.68 | 3.81 |
| | FX | 9.79 | 0.00 | 15.89 | 64.53 | 17.74 | FX | 12.34 | 0.24 | 0.27 | 4.23 | 16.02 | 5.32 | | | | | | | | | | | | | |
| Pa4 | FF | 12.53 | 0.00 | 14.95 | 62.20 | 15.96 | FF | 11.68 | 0.27 | 0.36 | 3.43 | 13.20 | 4.52 | 38.742 | −0.857 | $T_{-1}$ | 2.47 | 10.03 | 2.98 | 30.00 | −3.56 | 0.35 | 11.55 | 3.64 | 12.04 | 3.59 |
| | FX | 9.88 | 0.00 | 14.82 | 61.89 | 16.62 | FX | 11.89 | 0.26 | 0.36 | 3.41 | 13.19 | 4.53 | | | | | | | | | | | | | |
| Pa5 | FF | 8.73 | 0.00 | 16.79 | 63.46 | 18.53 | FF | 12.55 | 0.26 | 0.28 | 5.09 | 18.61 | 6.09 | 43.341 | −1.1047 | $T_{-2}$ | 3.50 | 14.23 | 4.29 | 6.72 | 9.94 | 0.65 | 27.45 | 3.50 | 14.23 | 4.29 |
| | FX | 7.62 | 0.00 | 16.74 | 63.12 | 19.07 | FX | 12.76 | 0.25 | 0.27 | 5.08 | 18.60 | 6.12 | | | | | | | | | | | | | |
| Po6 | FF | 6.27 | 0.00 | 10.14 | 50.70 | 11.94 | FF | 8.27 | 0.35 | 0.64 | 2.56 | 12.22 | 3.62 | 8.491 | 0.226 | $H_{-1}$ | 2.39 | 12.01 | 3.30 | 38.42 | −10.70 | 0.32 | 10.42 | 2.72 | 13.65 | 3.64 |
| | FX | 4.45 | 0.00 | 10.15 | 50.54 | 13.54 | FX | 8.45 | 0.34 | 0.65 | 2.53 | 12.15 | 3.61 | | | | | | | | | | | | | |
| Py-pa7 | FF | 6.69 | 0.00 | 12.23 | 55.56 | 14.25 | FF | 10.51 | 0.28 | 0.51 | 2.79 | 12.33 | 3.56 | 12.245 | 0.184 | $H_{-1}$ | 2.59 | 12.37 | 3.21 | 23.46 | −1.64 | 0.53 | 18.66 | 2.26 | 10.69 | 2.85 |
| | FX | 5.65 | 0.00 | 12.09 | 54.75 | 14.91 | FX | 10.66 | 0.27 | 0.52 | 2.76 | 12.21 | 3.53 | | | | | | | | | | | | | |
| Py-pa8 | FF | 3.98 | 0.00 | 9.86 | 49.94 | 13.10 | FF | 8.09 | 0.35 | 0.62 | 2.50 | 12.56 | 3.47 | 9.593 | 0.619 | $H_{-1}$ | 4.39 | 23.57 | 5.43 | 33.58 | −8.61 | 0.42 | 13.76 | 2.16 | 11.54 | 2.82 |
| | FX | 2.78 | 0.01 | 10.19 | 50.65 | 14.80 | FX | 8.28 | 0.33 | 0.62 | 2.46 | 12.36 | 3.45 | | | | | | | | | | | | | |

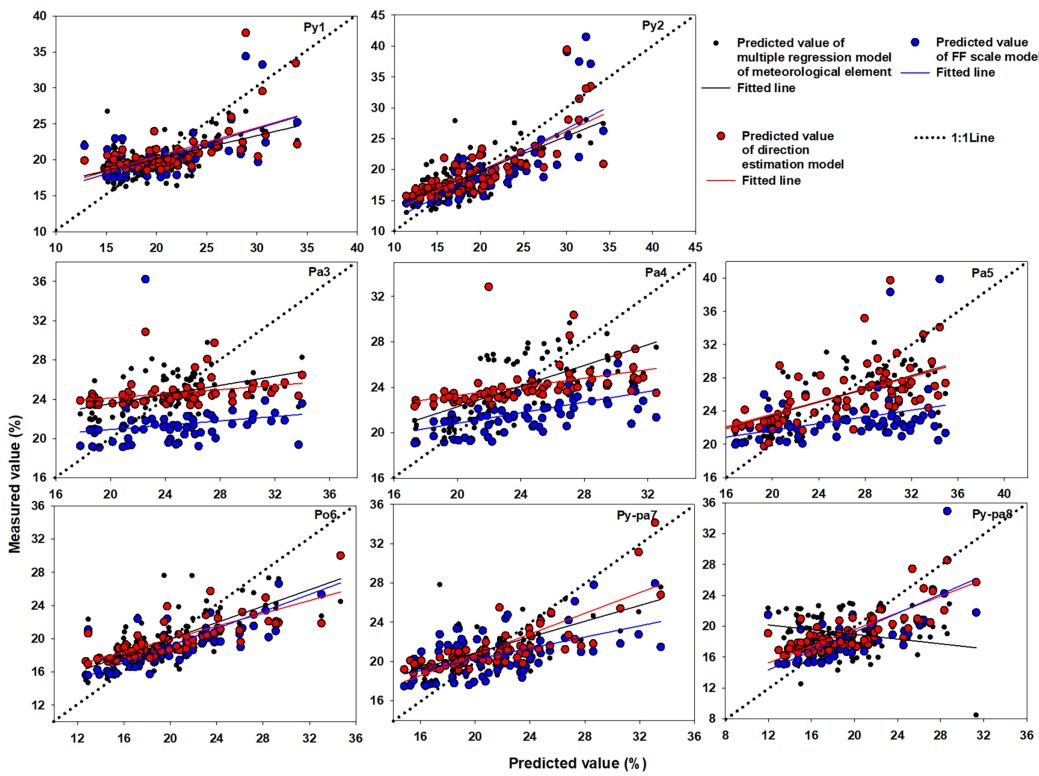

**Figure 3 Comparison between measured and predicted values.**

### Direct estimation model

When using the direct estimation method to predict the LMC, the minimum MAE was only 2.16%, and the maximum was 3.64%. The ranges of MRE and RMSE were 10.69–14.23% and 2.82–4.29%, respectively. The direct estimation method predicted the model best in Py-pa8, while Pa4 had the worst predictive effect. If the MRE limit is 15%, the LMC prediction results of all plots can be used for fire risk forecasting (Table 4).

## Model accuracy comparison

It can be seen that no significant difference was found in the prediction results of the moisture code regression models based on the FF and FX scale models, and overall, the predictions obtained based on the FF scale model were slightly better than those obtained based on the FX scale model. Therefore, regarding the comparison of prediction methods in this section, only comparisons between the moisture code regression method based on the FF scale model and the other two prediction methods were carried out.

### 1:1 comparison between measured and predicted values

Figure 3 shows a 1:1 comparison of the measured and predicted values of eight plots based on three LMC prediction models. With the exception of the litter in the Py-pa8 plot, in the remaining plots, when the measured LMC value was low, the values predicted by the three prediction models were lower than the measured value. As the measured LMC

values increased, the predicted values became higher than the measured values. In almost all plots, the straight line fitted by the direct estimation method was the closest to the 1:1 line, followed by that of the multiple regression method of meteorological element; the moisture code was the worst.

### Comparison of measured and predicted values

For the Py1 and Py2 litters, the direct estimation method and the moisture code regression method predicted value fluctuations that were close to the actual measured values, and the multiple regression method of meteorological element had a poor prediction effect when the LMC fluctuated frequently. For the litters of Pa3, Pa4 and Pa5, although the predicted values of the three prediction models showed similar fluctuations to the actual measured values to a certain extent, they were all in poor agreement. Among the prediction methods, the direct estimation method had a relatively good prediction effect, followed by the multiple regression method of meteorological element; the moisture code regression model prediction had the worst effect. For the litter of Po6, the predicted values of the three models were in good agreement with the actual measured values. For the Py-pa7 and Py-pa8 litters, the direct estimation method obtained the closest fluctuation between the predicted and the measured values, followed by the moisture code regression method, while the multiple regression method of meteorological element was poor (Fig. 4).

## Model extrapolation ability analysis

### Meteorological element regression model

When the meteorological element regression model was extrapolated, the minimum MAE was only 2.52%, consistent with the calculation results of the Po6 litter obtained using the Py2 litter prediction model. The maximum MAE was 8.80%, which appeared in the calculation results of the Py-pa8 prediction model for the Pa5 litter. The mean MAE value extrapolated by the meteorological element regression model was 4.61%, and the mean value of the coefficient of variation (CV) in the MAE was 0.32. The minimum and maximum MRE values extrapolated by the model were 12.24% and 51.36%, respectively. The locations of occurrence are the results of the Py1 prediction model for the Py-pa7 litter and the results of the Pa5 prediction model for the Py-pa8 litter. The mean value of MRE extrapolated by the model was 21.99%, and the coefficient of variation was 0.31 (Fig. 5).

### Moisture content regression model

Figure 6 shows the extrapolation of the moisture code model. It can be seen that the minimum extrapolated MAE was 2.55%, and the litter that appeared in Py-pa8 was calculated using the Po6 litter prediction model; the maximum MAE was 8.34%, which appeared in the litter of Pa5 and was calculated using the Py2 prediction model. The moisture code model extrapolated the average MAE to 4.11%, and the average coefficient of variation (CV) in the MAE was 0.35. The minimum and maximum MRE

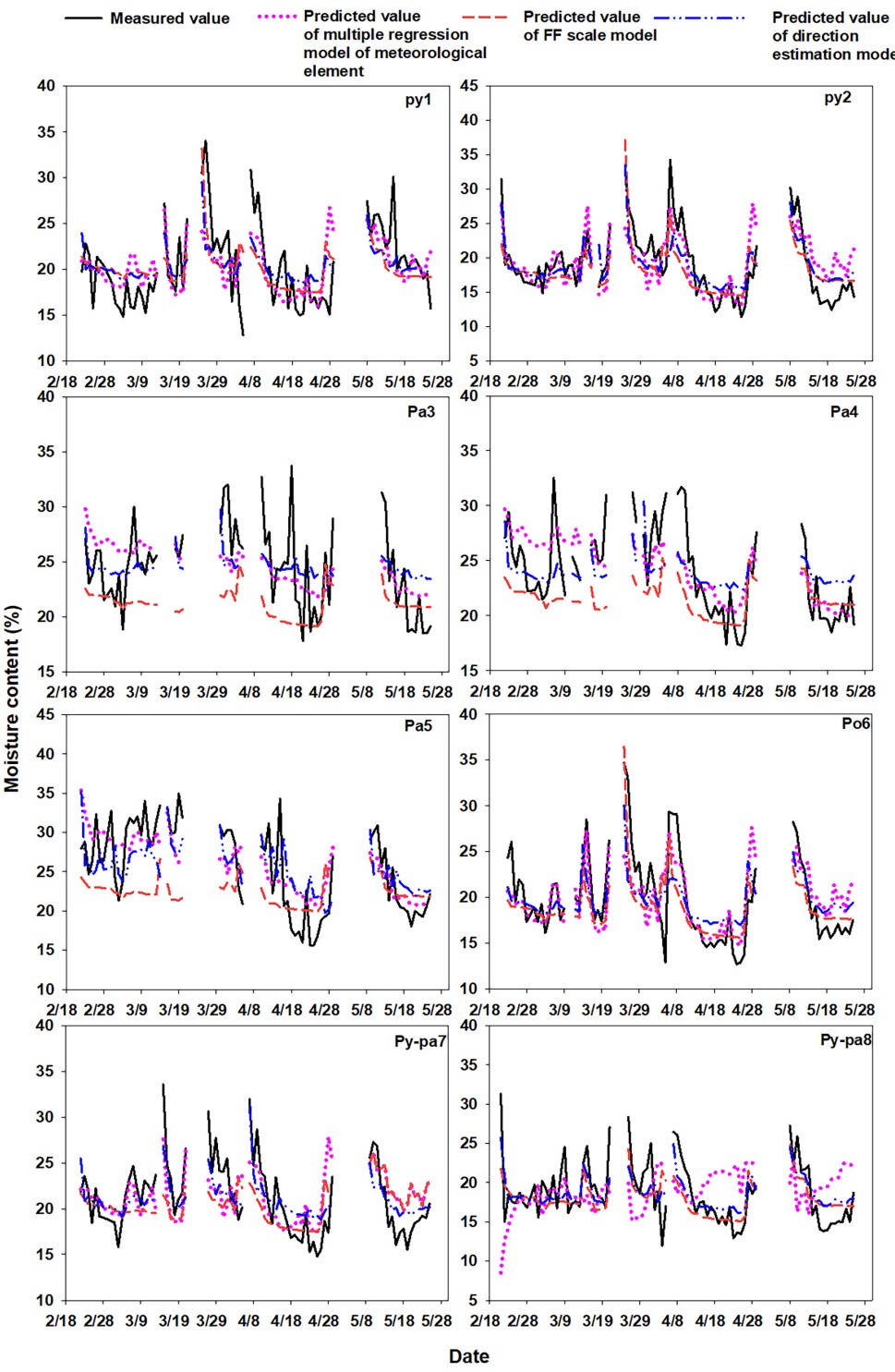

**Figure 4 Comparison of the measured and predicted value fluctuations during the monitoring period.**

 

Peer J

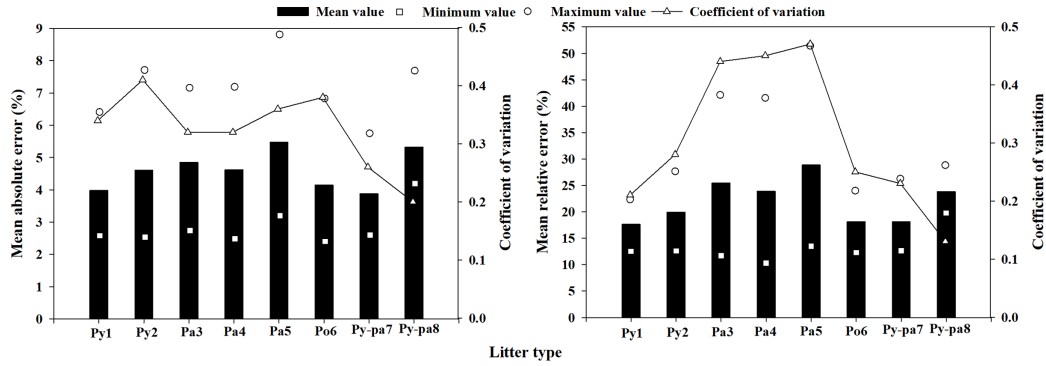

**Figure 5 Multiple regression model of meteorological element extrapolation errors and coefficient of variance in the eight plots.**

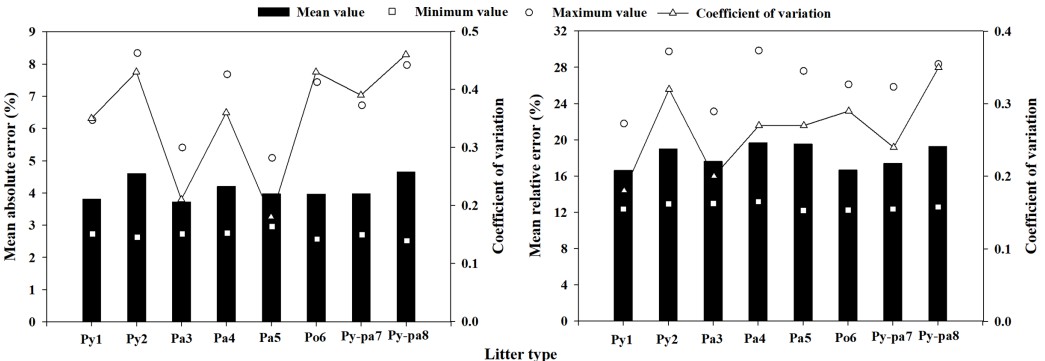

**Figure 6 Moisture code regression model extrapolation errors and coefficient of variance in the eight plots.**

values extrapolated by the moisture code model were 12.17% and 29.84%, respectively. The locations of occurrence resulted from the Pa4 prediction model for the Pa4 litter and from the Pa4 prediction model for the Pa3 litter. The mean MRE value extrapolated from the model was 18.22%, and the coefficient of variation was 0.27.

### Direct estimation model

The minimum MAE obtained when the model was extrapolated was 2.15%, which appeared in the Py-pa8 litter using the Py2 litter prediction model; the maximum MAE was 6.18%, which appeared in the Pa5 litter using the Py-pa8 prediction model. The direct estimation model extrapolated the average MAE to 3.72%, and the average coefficient of variation (CV) in the MAE was 0.26. The minimum and maximum MRE values extrapolated by the model were 11.23% and 30.99%, respectively, and the positions of occurrence are the results of the Py-pa7 litter prediction obtained using the Py1 prediction model and the Py2 litter prediction obtained using the Py1 prediction model. The mean MRE value extrapolated from the model was 17.19%, and the coefficient of variation was 0.23 (Fig. 7).
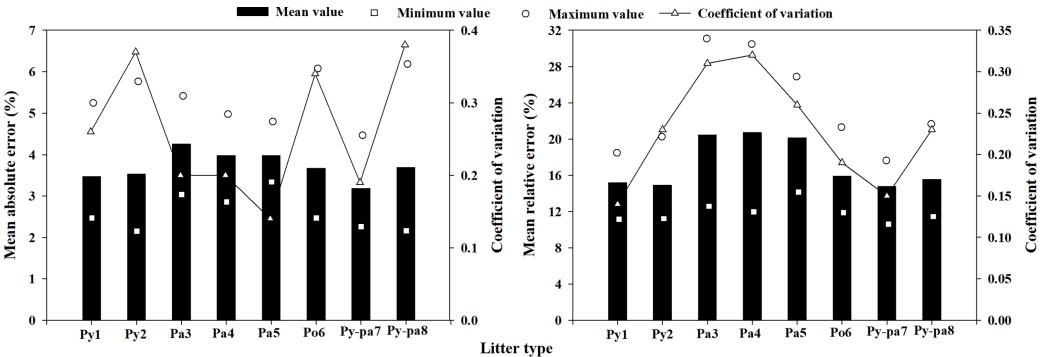

**Figure 7 Direct estimation model extrapolation errors and coefficient of variance in the eight plots.**

## DISCUSSION

### Difference and correlation analysis of LMC

There are differences in litter moisture content in different sampling areas, the main reason is that due to the different geographical locations (slope, slope direction, *etc*.), the microclimate conditions of litter are different, which affects the dynamic change of LMC (*Samran, Woodard & Rothwell, 1995*; *Keith, Johnson & Valeo, 2010*). In addition, different types of litter have different physical and chemical properties, resulting in different response to meteorological elements (*King & Linton, 1963*).

The dynamic changes in the typical litter moisture content in Yunnan Province have certain lags in their responses to all meteorological elements, and with increasing meteorological elements and sampling intervals, the significance first increases and then decreases. The results of this study are the same as those of Pixton and Warburton (*Pixton & Warburton, 1971*; *Britton et al., 1973*; *Zhang & Sun, 2020*). The main reason for this is that the dynamic change of LMC is not completely affected by the meteorological elements of the current day of the previous day, but mainly depends on the degree of its response to the meteorological elements. Therefore, the strongest correlation occurs from the current day to the previous *n* days. As the distance from the sampling time exceeds the day with the strongest correlation, the impact of meteorological elements on LMC decrease gradually. Therefore, there will be a significant increase first and then decrease.

The dynamic change of LMC in the study area is significantly negatively correlated with air temperature, and it increases with the increase of relative humidity, which is mainly due to the increase of temperature will improve the ability of air to carry moisture, which will increase the saturation humidity and decrease the relative humidity, the moisture of the litter will diffuse outward, and the LMC will decrease (*Alves et al., 2009*). The dynamic changes in the LMC of Pa3 and Pa4 had no significant relationship with wind speed; this result may have been caused by the excessive canopy closure of the two plots and the lower wind speeds in the forests (*Holdsworth & Uhl, 1997*; *Sun, Yu & Jin, 2015*). This study only analyzed the situations in which the LMC was lower than 35%,

and the dynamic change of LMC is less affected by rainfall. If all the moisture content data are considered, there is a significant correlation between the rainfall and the dynamic changes of LMC ($p < 0.001$). The accumulated rainfall in the first 3–7 days had a significant impact on the dynamic changes in LMC, while the accumulated rainfall in the first 1–2 days and the same day had no significant impacts, indicating that in the study area, the LMC could be reduced to less than 35% after 3–7 days of rainfall.

## Scale model applicability analysis

There was a significant difference between the moisture content values obtained by directly using the FF or FX scale models and the LMC values measured in all plots ($p < 0.001$). The mean error ranges of the eight plots obtained based on the FF and FX scale models were 5.59–14.20% and 4.30–13.72%, respectively; thus, the models all underestimated the measured values, similar to the results of *Zhang (2018)*. *Anderson & Anderson (2009)* took the gross moisture content as the research object and, based on the FF and FX scale models, obtained MEs of −11.81% and 7.52%, respectively. The FF scale model overestimated the measured values, and the FX scale model underestimated the measured values; these results may have been related to the LMC interval studied. In this study, only moisture contents below 35% were selected. During the study period, there was almost no rainfall, and the water loss rate under rainfall conditions in the scale model was not applicable and would cause the LMC values to be overestimated (*Zhang et al., 2017*). Therefore, the results of the two studies were different. Many studies have shown that direct use of the FFMC scale model is not applicable for LMC predictions, mainly due to the large gap between the litter type or stand structure and the actual situation at the sample site in the initial scale model study (*Chrosciewicz, 1989*; *Abbott et al., 2007*). For example, the scale model is obtained based on a study on litters with a thickness of 1.2 cm, while the thicknesses of the litters in most sample areas examined in this study were above 3.0 cm. In addition, the key parameters that represent set values such as the equilibrium moisture content and time lag in the model cause great errors when compared with the actual situation; thus, this model it cannot be directly used.

## Model predictors and parameters

In the meteorological element regression model, the predictive factors are the same among the same litter type and different places, indicating that the LMC response to meteorological elements is better than the response to forest stand conditions due to the influence of the litter type. Except for the Pa litter, relative humidity was the predictor for all the other LMC meteorological element models; this result further proves that relative humidity has the greatest influence on dynamic changes in the LMC (*Pixton & Warburton, 1971*; *Jiménez et al., 2016*). The stepwise regression method is selected to establish the multiple regression model of meteorological elements, the most important variables are selected from a large number of selectable meteorological element as the predictor variable of the model, which avoids the collinearity between meteorological elements. Therefore, some litter prediction models choose relative humidity, and some are

temperature, but they are essentially multiple regression models of meteorological elements of LMC, so they are comparable to each other.

The parameters α and β of the FF moisture code regression models of the 8 plots ranged from 7.21-12.55 and from 0.25–0.39, respectively; in the FX moisture code regression model, these parameters ranged from 7.40–12.76 and from 0.25–0.37, respectively. The α values (1.9893 and 4.6381) in the FF and FX moisture code regression models obtained by Anderson were lower than those obtained in this study, and the β values (0.6545 and 0.5868) of Anderson were higher than those obtained in this study (*Anderson & Anderson, 2009*). This is mainly because the LMC data selected for analysis in this study had values less than 35%, while Anderson used all LMC data.

The minimum value of the parameter α in the LMC prediction model of the direct estimation method was 6.72, and the maximum value was 37.70. The results of *Catchpole et al. (2001)* study found that the range of α was 0.26–0.37 (*Catchpole et al., 2001*). *Sun, Yu & Jin (2015)* classified typical Daxinganling forests as the research objects, and the results showed that the range of α changed from 0.087–0.594. *Zhang & Sun (2020)* took *Quercus mongolica* and Korean pine litter as the research objects and obtained α values of 0.25 and 0.004, respectively. The results of this study were significantly higher than those of other scholars, mainly due to the different research step lengths used. The monitoring step length of this study was 24 h. If the monitoring step length increased, α would increase to a certain extent. The absolute value of β in the Nelson model represents the response degrees of dynamic changes in the LMC to environmental elements. A larger absolute value of β indicates a more sensitive responses of dynamic changes in the LMC to environmental elements (*Nelson & Ralph, 1984*). For the same type of litter, the absolute value of β in the LMC prediction model is higher for sunny slopes than for shaded slopes. This is similar to the results of *Sun, Yu & Jin (2015)*. The dynamic changes in the LMC observed on sunny slopes are more sensitive to the response of meteorological elements than those observed on shaded slopes. Because the physical and chemical properties of litter and the bed structure have significant effects on dynamic changes in the LMC, the β values differ among different litters.

### Time lag of litter bed

The direct estimation method shows that the time lag of the litter beds in 8 plots ranged from 10.42–27.45 h; these values were higher than those reported in the results of *Catchpole et al. (2001)*. Because of the different physical and chemical properties of litter, the water diffusion rates also differ. In addition, these results are also related to experimental methods. Catchpole spread litters on the land surface, and water vapor exchange occurred almost exclusively between the litter and the external environment. The research object of this paper was the litter bed, and the moisture change rate was analyzed in addition to the responses of the litters to the external environments; these factors were also related to the internal structure of the litter bed and the water vapor exchange between the litter and the litter in the bed (*Liu, 2007*; *Ruiz, Vega & Álvarez, 2009*; *Jin & Chen, 2012*). The average thickness of the litter beds examined in this study was 3.5 cm, which was higher than those analyzed other studies. Therefore, the change rate of

water obtained in this study was lower than that of *Catchpole et al. (2001)* with a higher time lag.

## Model accuracy analysis

The mean MAE and MRE values of the multiple regression methods of meteorological element applied for the 8 sample plots in this study were 2.97% and 14.06%, respectively. *Lu (2016)* used the typical litter in the Nanweng River as the research object and established a regression model of meteorological elements. The minimum MAE and MRE values of their model were as high as 18.9% and 27.7%, respectively; these values were significantly higher than those obtained in this study. The main reason for this discrepancy is that this study only conducted research on data in which the LMC values were below 35%, and more predictive factors were considered entering the models, making the models more comprehensive. Therefore, the regression model error of the meteorological elements with a daily step size was lower. *Sun, Yu & Jin (2015)* took the typical litter in the Daxinganling Mountains as the research object and obtained MAE and MRE values of the LMC meteorological element regression model of 1.48% and 9.01%, respectively. *Zhang & Sun (2020)* obtained LMC meteorological element regression model errors of *Quercus mongolica* and Korean pine of 1.95% and 13.26%, respectively. The model errors obtained in these studies were slightly lower than those obtained in this study, mainly due to the different research steps; the previous studies all used hours as the step length.

The mean MAE of LMC values obtained from the moisture code regression models based on the FF and FX scale models for the eight plots were 3.27% and 3.26%, respectively, and the MRE means were 14.04% and 14.07%, respectively; these values were similar to the research results of *Anderson & Anderson (2009)*. The prediction error of the scale regression model was higher than that of the meteorological element regression model. This may be because the scale model itself was not applicable, and its built-in equilibrium moisture content, time lag and other key parameter models were also not applicable to the typical litter in Yunnan (*Anderson & Anderson, 2010*; *Zhang, 2018*). In this study, the litter bed structure, physical and chemical properties, geographical conditions and climate zone were significantly different from the FWI system study area and litter types, so this model may not be applicable (*Abbott et al., 2007*; *Schiks & Wotton, 2014*).

The mean MAE and MRE values of the LMC prediction models of the eight plots obtained by the direct estimation method were 2.82% and 12.76%, respectively. These results are lower than those reported in the study of *Lu (2016)*. Although the study step was the same between these two studies, this study only considered LMC data with values less than 35% instead of rainfall data, thus improving the prediction accuracy, so the prediction accuracy of this study was higher than the results of *Lu (2016)*. *Catchpole et al. (2001)* chose the direct estimation method to obtain MAEs of the LMC prediction model with a range of 0.8–1.9%. The MAEs of *Quercus mongolica* and *Pinus koraiensis* obtained by *Zhang & Sun (2020)* were 1.02% and 4.73%, respectively. The prediction

effects were better than in their study than those obtained this study, which was also caused by the prediction step.

Through the *t*-test, no significant difference was observed among the MAEs (MREs) of the three LMC prediction methods, but the direct estimation method had the lowest error and the best prediction effect.

## Model extrapolation ability analysis

The extrapolated average MAEs of the multiple regression method of meteorological element, scale model regression method and direct estimation method were 4.61%, 4.11% and 3.72%, respectively, and the MREs were 21.99%, 18.22% and 17.19%, respectively. The extrapolation error of the multiple regression method of meteorological element was significantly higher than those of the other two methods, while the moisture code regression method and the direct estimation method had no significant difference. In addition, the direct estimation method had the smallest mean coefficient of variation value, at only 0.26, which was lower than those of the other two methods. Therefore, the best extrapolation effect was the direct estimation method, followed by the moisture code regression method, while the worst was the multiple regression method of meteorological element; these results are consistent with the research results of *Yu, Jin & Di (2013a)*, *Yu, Jin & Di (2013b)*. Consistent with the results reported in section **model accuracy analysis**, because the equilibrium moisture content and time lag model were not corrected in the FFMC, the moisture code regression method was not applicable for predicting the litter moisture content in this area, and the prediction error was large. The LMC prediction model obtained by the multiple regression method of meteorological element had a poor extrapolation ability, and some extrapolated MREs were as high as 50% or more; this model cannot meet the daily fire risk forecast demand. The direct estimation method can be used to predict the typical LMC in Yunnan Province, and the predictive ability and extrapolation ability of this method can fully meet the daily fire risk forecast demand.

## CONCLUSIONS

The results showed that the dynamic change in the typical litter moisture content in Yunnan Province had an obvious lag under different stand conditions; these variations were mainly related to the air temperature, humidity and wind speed. With an increase in meteorological elements and the sampling interval, the significance first increased and then decreased. Direct use of the scale model for LMC predictions was not applicable because there was a significant difference between the predicted and the measured values. Although the prediction model obtained by the regression had a good extrapolation ability, it also had a large error. The prediction error of the multiple regression method of meteorological element was also large, and its extrapolation ability was poor. Therefore, neither of these two methods could meet the daily fire risk forecast accuracy requirements of the study area. The prediction accuracy and extrapolation ability of the direct estimation method obtained based on the Nelson equilibrium moisture content equation could meet the daily fire risk forecast in this area.

The research on the dynamic changes in LMC and the prediction models conducted in this study aimed to lay a foundation for local daily fire risk forecast research. If the LMC prediction values are applied to predict fire behaviors, the accuracy of this research would not be satisfied, and it would be necessary to further shorten the LMC research scale and improve the forecast accuracy. This study also has some limitations. For example, the litter bed structure, especially the litter bed density and composition, has a certain impact on the dynamic changes in the LMC and on the prediction results. In addition, the selection of the equilibrium moisture content equation in the direct estimation method also impacts the prediction results, and this study did not consider the dynamic changes of LMC under extreme climates. In future research, the litter bed structure should be considered comprehensively, the research step length should be shortened, the influence of extreme climates on the LMC needs to be considered and more prediction methods of LMC also need to be considered (*Matthews, 2014*; *Resco de Dios et al., 2015*). These advancements will provide great significance for understanding the dynamic change mechanism of the LMC and for improving fire prediction accuracy.

## ACKNOWLEDGEMENTS

We are indebted to the graduate students Yong Zhou, Jinbo Liu for their support in field work. We are also grateful to two anonymous reviewers for their valuable comments.

### Funding

This work was supported by the Natural Science Foundation of Guizhou Province grant number ZK [2021] general 158 and the Guizhou provincial first-class major (biological science) project (Education Department of Guizhou province) [2019] 46. The funders had no role in study design, data collection and analysis, decision to publish, or preparation of the manuscript.

### Grant Disclosures

The following grant information was disclosed by the authors:
Natural Science Foundation: ZK [2021] general 158.
Education Department of Guizhou province: [2019] 46.

### Competing Interests

The authors declare that they have no competing interests.

### Author Contributions

- Yunlin Zhang conceived and designed the experiments, analyzed the data, prepared figures and/or tables, authored or reviewed drafts of the paper, and approved the final draft.
- Lingling Tian performed the experiments, prepared figures and/or tables, and approved the final draft.

## Data Availability

The raw data is available in the Supplemental File.

## Supplemental Information

Supplemental information for this article can be found online at http://dx.doi.org/10.7717/peerj.12206#supplemental-information.

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
