# Peer review of "Dynamic changes in moisture content and applicability analysis of a typical litter prediction model in Yunnan Province"

_PeerJ, doi:10.7717/peerj.12206_

## Round 0.1 · original submission · Major Revisions

We now have received three review reports on your manuscript. I have considered them, and based on the advice from the reviewers I think that your manuscript should be revised according to the review comments before its publication.

Reviewer 1 ·

Basic reporting

Forest fire risk prediction is the most important task of forest protection, and the litter moisture content directly affects the occurrence probability of forest fire and a series of fire behavior characteristics after being ignited. The dynamic change and prediction model of litter moisture content is the core content of forest fire prediction. The wet and dry seasons are distinct in Yunnan Province, and the precipitation is scarce during the dry season, the risk of forest fire occurrence is very high. The results of this manuscript have reference value for the forest fire prevention and prediction in dry season and formulation fire prevention strategies in Yunnan Province. The structure, figures, tables are well organized, and the findings are interesting, but more discussion needed. It is particularly necessary to discuss why the litter moisture content is mainly related to the air temperature, relative humidity, and wind speed, but weakly correlated with precipitation. Overall, the manuscript needs further modification before it can be accepted for publication.

Experimental design

no comment

Validity of the findings

no comment

Additional comments

1. Overview of the study area: It would be better to add a detailed study area map in the section.
2. Sample plot setting: Among the four types of forest land, why the number of sample plots in each forest type is different, and the slope position and depth are also different? The criteria and reasons for selecting sample plots need to be explained.
3. In the section of meteorological elements regression model, some sample plots selected relative humidity, while other sample plots use air temperature as the predictor variable. What are the reasons for choosing different predictor variable, and are the results of different predictor variable comparable?
4. Line 223–224, “…… when the LMC value is lower than 33%, it is difficult for forest fires to occur……”, is it not that the lower the litter moisture content, the easier it is to ignite?
5. Section 2.3.1 is mentioned many times in the paper, but the contents of each part are not numbered.
6. The citation format is incorrect in Line 83, 429–431.
7. Why is the broken line not continuous in Figure 3?
8. It is better to use double brackets instead of single bracket for formula numbers.
9. In the note of Table 1, P5 should be Pa5.

Reviewer 2 ·

Basic reporting

no comment

Experimental design

Describtion of methods should be improved with more deatils, such as the reason of choosing these meteorological elements to build regression model, the reason of choose the Nelson model, and the reason of choosing the stepwise regression should be added.

Validity of the findings

no comment

Additional comments

This article combined multiple models to predict the litter moisture content, which can provide a certain practical reference for prevention of fire occurrence. The article has reached the requirements of the journal in terms of structure and representation. However, improvements are still needed in many places, and it is recommended to consider the manuscript after further revisions

Annotated reviews are not available for download in order to protect the identity of reviewers who chose to remain anonymous.

Reviewer 3 ·

Basic reporting

no comment

Experimental design

no comment

Validity of the findings

no comment

Additional comments

I started reading the manuscript from Zhang and Tian. It covers an important problem or an area where we are lacking data. However, the manuscript needs a lot of effort. I recommend the authors to seek assistance from a colleague with experience in writing scientific articles. This manuscript needs English editing but also help with the overall presentation of the results. Asking a reviewer for detailed input on this is too much, as this manuscript is actually a draft. The data is interesting and, if they can manage to solve this formal aspects, it may deserve eventual acceptance. It is very far from that stage right now.

Specific comments.
L66 - I disagree with this view. See (Nolan et al. 2016) for instance

L122 - indicate here number of plots and summary of main characteristics
L125 - what did you actually sample? provide indication on particle size, type of material and whether it was suspended o profile.
L135 - if there was water on the samples (rain, dew, etc), data is not valid.
Eq. 1 - delete "%"
L147 - at which height?
L150 - provide summary here
L152 - this is multiple regression, not "element" regression
Table 3 - what is litter type?

Fig. 2 shows that none of the models performed well. Try other methods like (Resco de Dios et al. 2015), Sharples, etc

Tables 3-7 are not necessary. Just present a single table with validation statistics for all models.



Nolan, R.H., V. Resco de Dios, M.M. Boer, G. Caccamo, M.L. Goulden and R.A. Bradstock. 2016. Predicting dead fine fuel moisture at regional scales using vapour pressure deficit from MODIS and gridded weather data. Remote Sensing of Environment. 174:100-108.
Resco de Dios, V., A.W. Fellows, R.H. Nolan, M.M. Boer, R.A. Bradstock, F. Domingo and M.L. Goulden. 2015. A semi-mechanistic model for predicting the moisture content of fine litter Agricultural and Forest Meteorology. 203:64-73.

---

## Round 0.2 · Minor Revisions

Before your paper is accepted for publication, I suggest that you revise your manuscript again according to the review comments.

Reviewer 1 ·

Basic reporting

The authors have addressed all my comments adequately and the paper can be accepted. But there are still some spelling errors and common sense questions, please check and modify them carefully.

Experimental design

no comment

Validity of the findings

no comment

Additional comments

The authors have addressed all my comments adequately and the paper can be accepted. But there are still some minor questions, please check and modify them carefully. There are many comments as follows:
1. L162: This is meteorological elements, not meteorological the elements.
2. In a closed environment, the air atmospheric pressure increases with the increase of air temperature. In an open environment, the situation is opposite. In Line178-179: “The increase of air temperature will lead to the increase of air atmospheric pressure……”. Is your research in a closed environment? There are similar expressions in the following text, please check and modify it carefully.
3. L198: The word “inteoduced” should be change to “introduced”.
4. L618: The first letter of “these” should be capitalized.

Reviewer 2 ·

Basic reporting

no comment

Experimental design

no comment

Validity of the findings

no comment

---

## Round 0.3 · accepted · Accept

After reviewing your revised manuscript of the current version (version 2), I decided to accept it for publication in PeerJ.